# Electrocaloric cooling system utilizing latent heat transfer for high power density
Julius Metzdorf [1] ✉, Patrick Corhan[1], David Bach [1], Sakyo Hirose [2], Dirk Lellinger [3], Stefan Mönch [4], Frank Kühnemann [1], Olaf Schäfer-Welsen[1] & Kilian Bartholomé [1] ✉

Electrocalorics (EC) is potentially more efficient than refrigeration and heat pumps based on compressors and does not need detrimental fluids. Current EC-prototypes use solid-state contact or forced convection with liquids to transfer the heat generated from the EC-material, which inhibits high cycle frequencies and thus limits power density. Here we present a heatpipe system solution, where the heat transfer is realized through condensation and evaporation of ethanol as a heat transfer fluid. Our prototype with lead scandium tantalate (PST) EC-material working at 5 Hz shows a specific cooling power of 1.5 W g$^{-1}$. This is one order of magnitude more than previously reported for ceramic EC-prototypes. Overcoming the limits of slow heat transfer is essential to reach high specific cooling powers enabling a future commercial success of the technology.

The sector of cooling and air conditioning consumes more energy each year and it is expected to add as much as the current electricity demand of the European Union by 2050 to the worldwide demand, if no political or technological improvements are made[1]. Therefore, the efficiency of these systems becomes more important. Today the vapor-compression refrigeration is the state-of-the-art technology. However, most compressors reach efficiencies below 50% of the thermodynamically possible Carnot limit[2]. Additionally the hydrofluorocarbons, which are frequently used as refrigerants, have a strong greenhouse potential[3]. Their use in the European Union and around the world is regulated to be reduced strongly in the next years[4]. But also, alternative refrigerants can have disadvantages: they can be toxic, highly flammable or high-pressure fluids.

Caloric cooling systems are a promising alternative to the vapor-compression technology, as they can work without harmful fluids. Caloric materials exhibit a reversible temperature change when a field is applied (magnetic, electric, or mechanical stress). The heat can be transferred to a heat sink. When the field is removed, the caloric material cools down and thermal energy can be absorbed from a heat source. By cyclic repetition of these steps, heat can be transferred from a cold to a hot reservoir.

Electrocaloric-devices are especially interesting for the following reasons: The generation of electrical fields is rather easy and low cost, compared to the generation of strong magnetic fields or mechanical stress. Using electricity as driving force also has other advantages: A fully electric energy conversion is sufficient, and no additional energy conversion from electrical to magnetic or mechanical energy is necessary (as in magnetocaloric or

mechanocaloric devices). And almost all work to drive the EC-components can be recovered[5,6].

The EC-prototypes built until today can be classified by their heat transfer mechanism[7]: Prototypes, which use a fluid for heat transfer and those, which use solid state contact. Using a fluid for heat transfer and a regenerator is a well-known concept from magnetocaloric research. With this active electrocaloric regenerator concept Torello et al.[8] reached a temperature span of 13 K. However, the specific cooling power (measured with a similar device by the same group) was only 0.012 W g$^{-1}$ (limited by the low cycle frequency of 0.08 Hz) As the fluid itself has some thermal capacity, which must be warmed and cooled by the EC-components, mostly multi-layer ceramic capacitors are used for this kind of prototypes. On the other hand, there are prototypes with solid state contact as heat transfer mechanism. Most of these prototypes use cascading of several EC-components to leverage the adiabatic temperature change of the material. The most performant prototypes use the flexibility and electrostatic response of polymer EC-components to switch the thermal contact between hot and cold side (or the next stage). A four-stage device showed a temperature span of 8.7 K at no load and a high specific cooling power of 2.140 W g$^{-1}$ (90 mW cm$^{-2}$) at 0 K temperature span and cycle frequency of 1 Hz. A COP/COP$_{Carnot}$ of 12% was calculated for 6 K temperature span[9]. The highest specific cooling power of 3.950 W g$^{-1}$ (with 4.6 K) temperature span under no load conditions at a cycle frequency of 1.5 Hz reported until today uses the same concept as before but employs a hybrid material made of a polymer with ceramic nanoparticles[10].

[1]Fraunhofer Institute for Physical Measurement Techniques IPM, Freiburg, Germany. [2]Murata Manufacturing Co., Ltd., Kyoto, Japan. [3]Fraunhofer Institute for Structural Durability and System Reliability LBF, Darmstadt, Germany. [4]Fraunhofer Institute for Applied Solid State Physics IAF, Freiburg, Germany. ✉e-mail: julius.metzdorf@ipm.fraunhofer.de; kilian.bartholome@ipm.fraunhofer.de

EC-materials are a topic of current research and especially new polymers and hybrid materials are reported in the last years[10–12]. It should be mentioned that polymers and hybrid materials can withstand higher electrical fields and show higher entropy change per mass[13] than ceramic EC-material. This means more thermal energy is pumped per cycle, which raises the specific cooling power.

Apart from high efficiencies and higher device temperature spans, high specific cooling powers are crucial for applications. Not only because it enables small devices, but also the device costs are reduced, when less EC-material is needed. The specific cooling power of any caloric device is not only determined by the thermal energy pumped per cycle, but also by the achievable cycle frequency.

In current electrocaloric systems the cycle frequency is below 1.5 Hz (see Supplementary Table 1) because of three reasons: The low surface to volume ratio (especially for multilayer ceramic capacitors), the low thermal conductivity of the EC-material itself and the employed heat transfer methods. Now we have a closer look at the heat transfer methods. When working with a liquid fluid, the thermal coupling of the EC material to the fluid is good. But the fluid velocities have to be low to prevent turbulences, which increases the hydraulic losses and deteriorate the established temperature gradient and system efficiency[7].

Working with solid state contact, the heat transfer coefficient strongly depends on the surface roughness and the contact pressure[14]. For ceramic as well for polymer EC-materials the contact pressure, which can be applied is rather limited. High contact pressures would need extra energy and elevate the risk of mechanical failure.

In this work, we present the active electrocaloric heatpipe (AEH) concept, which uses evaporation and condensation as a heat transfer mechanism. Thus, higher cycle frequencies and specific cooling powers can be reached. With ceramic $PbSc_{0.5}Ta_{0.5}O_3$ (lead scandium tantalate, PST) multilayer capacitors we reach a specific cooling power (per active mass) of $1.5\,W\,g^{-1}$, which is one order of magnitude higher than other prototypes based on ceramic EC-material. Although polymer or hybrid systems show higher specific power, our system shows an absolute cooling power of 2.2 W and a cooling power density (per active volume) of $13.5\,W\,cm^{-3}$, the highest reported values for an electrocaloric prototype. The concept of the AEH can overcome the substantial limitations of heat transfer observed in other prototype designs leading to high power densities and thus pave the way for EC cooling to be lightweight, compact and cost-competitive.

## Results and discussion

### The active electrocaloric heatpipe concept

High heat transfer coefficients can be obtained using the latent heat of condensation and evaporation. This concept is used successfully in heat pipes. A pure fluid is enclosed in a container without any other fluids. This leads to the evaporation of the liquid until the vapor pressure of the fluid temperature is reached. In this two-phase-region the vapor pressure is governed by the temperature according to the vapor pressure curve. Each temperature increase is followed by evaporation of liquid and results in a pressure increase. If heat is applied to one side of a heat pipe, liquid evaporates, pressure rises and subsequently vapor condenses on the cold side, releasing the enthalpy of evaporation, which results in a very efficient and fast heat transfer.

This concept of latent heat transfer has previously been reported for magneto[15]- and elastocaloric systems[16] and is by this work now extended to the active electrocaloric heatpipe (AEH). The basic unit of the AEH is one segment (within the heatpipe) consisting of EC-components, a heat transfer fluid and passive check valves, which function as thermal diodes, as shown in Fig. 1. Vapor can only flow through the check valves from the cold side to the hot side. A detailed description and characterization of the check valves is given by Maier et al.[17]. Whenever an electric field is applied to the EC-components, their temperature increases due to the electrocaloric effect. Thereby the liquid fluid evaporates from the surface of the EC-components, and the pressure inside the segment increases. This leads to an opening of the check valve and a gaseous fluid flow to the hot side, transferring heat

from the EC-components to the hot side (condenser). When subsequently the electric field is turned off, the EC-components cool down and gaseous fluid condenses on the components. This leads to a decrease in pressure in the segment, opening the check valve to the cold side and letting gaseous fluid enter from the evaporator. Thus, by an alternating electric field, a unidirectional gas flow from the cold side to the hot side occurs, whereby heat is pumped. The accumulating liquid in the condenser (hot side) flows back to the evaporator (cold side) through a wick. The working principle of the AEH is shown in more detail in the Supplementary Video. Due to unbalanced heat and mass fluxes, the system requires a thermal stabilization. This is incorporated by a local fluid return, which is fed by fluid of the condenser (see Supplementary Notes and Supplementary Fig. 1 - Supplementary Fig. 3 for a detailed description of the local fluid return).

Using 10 PST multilayer capacitors components, provided by Murata Manufacturing Co., Ltd. each with an active mass of 0.15 g a one-stage electrocaloric system was built (see Supplementary Fig. 2 for a schematic illustration of the setup, Supplementary Fig. 4 for a schematic cross-section of the EC-segment and Supplementary Fig. 5 for the experimental realization). The adiabatic temperature change $\Delta T_{ad}$ was determined to be 1.7 K at 29 °C at a field change of 5.2 V/µm (see Supplementary Fig. 6). The heat transfer fluid was ethanol. We wrapped the evaporator with a heating wire and insulated it against the environment. Applying electrical power to the heating wire, the heating load can be adjusted. The condenser temperature can be controlled via a thermal bath. For details of the setup see the methods section and the Supplementary Information.

### Specific cooling power and cut-off frequency

The system was characterized by measuring the temperature span between evaporator and condenser for different heating powers (Fig. 2). This was done also for different frequencies up to a maximum frequency of 5 Hz (Fig. 3 and Supplementary Fig. 7), resulting in a maximum cooling power of 2.2 W and a maximum temperature lift of 0.4 K.

From this data, the maximum specific cooling power $\dot{q}$ was determined and analyzed in dependence of the cycle frequency $f$ using the equations derived by Hess et al.[18]:

$$\dot{q}(f, \Delta T = 0) = \frac{\dot{q}_{\max}}{\sqrt{1 + \left(\frac{f}{f_c}\right)^2}} \frac{f}{f_c} \tag{1}$$

where $\dot{q}_{\max}$ is the maximum specific cooling power and $f_c$ is the cut-off frequency. This cut-off frequency is a measure at which cycle frequency the cooling power starts to saturate. To reach high specific cooling powers a high cut-off frequency is needed, which depends on the heat capacity $c_p$, the thermal resistance $R_{th}$ of the system and the mass $m$ of the EC-components:

$$f_c = \frac{1}{2\pi R_{th} c_p m} \tag{2}$$

The maximum reachable specific cooling power is given by:

$$\dot{q}_{\max} = \frac{\Delta T_{ad}}{2\pi R_{th} m} \tag{3}$$

where $\Delta T_{ad}$ is the adiabatic temperature change of the electrocaloric material.

We measured a specific cooling power of $1.5\,W\,g^{-1}$ at a system frequency of 5 Hz. Figure 4 shows the frequency dependency of the measured specific cooling power.

With the mass of the EC-components is $m = 1.5$ g, the heat capacity $c_P = 310\,J\,K^{-1}\,kg^{-1}$[19], and the adiabatic temperature change $\Delta T_{ad} = 1.7K$ (Supplementary Fig. 6), the only free parameter left is the thermal resistance $R_{th}$. Fitting equation (1), (2) and (3) to the data shown in Fig. 4, a value of $R_{th} = 0.09 \pm 0.04\,W\,K^{-1}$ is obtained. This

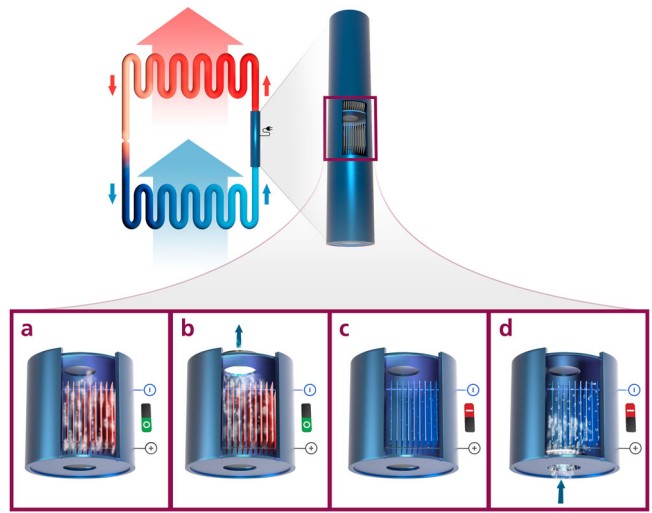

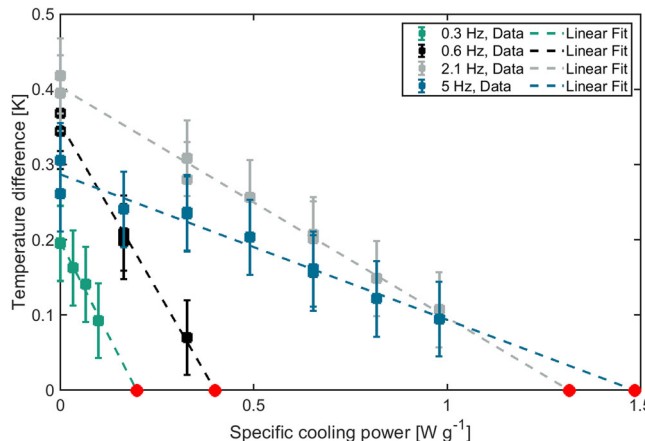

**Fig. 1 | Schematic working principle of the active electrocaloric heatpipe. a** The electric field is applied and the electrocaloric (EC) components heat up. Subsequently liquid evaporates. **b** The pressure increases and opens the upper check valve. The vapor flows to the next segment, transporting the latent heat of evaporation. **c** The electric field is removed: The EC-components cool down and vapor condenses on them. **d** Due to the condensation of vapor, the pressure is reduced, and the lower check valve opens to let in vapor from the segment below. Thus, a directed heat flow from bottom to top is achieved by using the latent heat of the fluid.

**Fig. 3 | Temperature span plotted against the specific cooling power for different frequencies.** The raw data is fitted with a linear regression for the different cycle frequencies. The interception of the fit with the x-axis ($\Delta T = 0$, marked with red dots) is used as the specific cooling power $\dot{q}$ for each frequency. The quantification of the error bars is described in detail in the methods section "Quantification of measurement uncertainties".

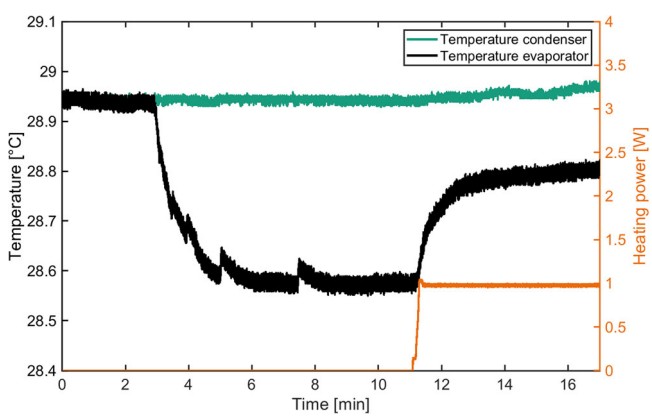

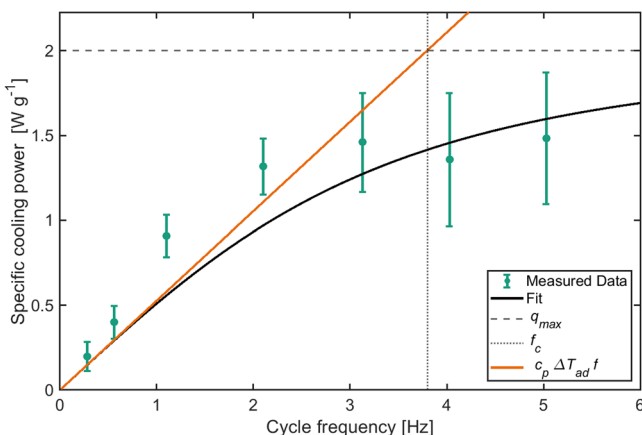

**Fig. 2 | Temporal evolution of the condenser and evaporator temperatures of our system.** After application of a sinusoidal voltage to the electrocaloric components a temperature difference establishes (no load condition). When the evaporator is heated, the established temperature difference reduces. For each heating power the steady state temperature difference is recorded (corresponding to one data point in Fig. 3). The amplitude of the sinusoidal heating power is shown in orange.

**Fig. 4 | Measured specific cooling power per active mass for zero temperature span (green circles) as a function of the cycle frequency.** Equation (1) was fitted to the data (black line), providing the maximum achievable specific cooling power (dashed line) and the cut-off frequency (dotted line). The orange line is the theoretical optimum, without thermal resistance. The quantification of the error bars is described in detail in the methods section "Quantification of measurement uncertainties".

gives a cut-off frequency $f_c = 3.8$ Hz and a maximum specific cooling power $\dot{q}_{max} = 2.0$ W g$^{-1}$.

This thermal resistance $R_{th} = 0.09 \pm 0.04$ W K$^{-1}$ includes the thermal conductance through the EC-components, the heat transfer to the fluid and the pressure drop across the check valves. The check valves have negligible thermal resistances as shown by Maier et al.[17].

The thermal resistance in our system mainly results from the low thermal conductivity of the EC-components combined with their geometry. We can see this by calculating the thermal diffusion length at the cut-off frequency given the thermal conductivity of the PST material $\lambda_{PST} = 1$ W m$^{-1}$ K$^{-1}$[20] and the density $\rho_{PST} = 9070$ kg m$^{-3}$[21].

$$l = \sqrt{\frac{\lambda_{PST}}{\pi f_c \rho_{PST} c_P}} = 173 \ \mu m \qquad (4)$$

This is roughly 40% of the thickness of the EC-components. This means that for frequencies around the cut-off frequency or higher the heat does not flow sufficiently fast from within the components to their surface.

## Comparison to literature

In Fig. 5 the specific cooling power (per active mass) obtained with the AEH is compared to the values reported in the literature.

It is noteworthy that the specific cooling power is one order of magnitude higher than for other prototypes working with ceramic EC-components. Jia & Sungtaek[22] did not present pure electrocaloric cooling, therefore their work was excluded here. In general EC polymers show higher entropy change per mass $\Delta s$[13], which results in higher specific cooling powers. But the AEH working with ceramic EC-components reaches a specific cooling power almost as high as the best prototypes working with polymers or hybrid materials.

Regarding absolute cooling power we reached 2.2 W, which is among the highest value ever reported for an EC-device. With 13.5 W cm$^{-3}$ we also reach the highest cooling power density per active volume (see

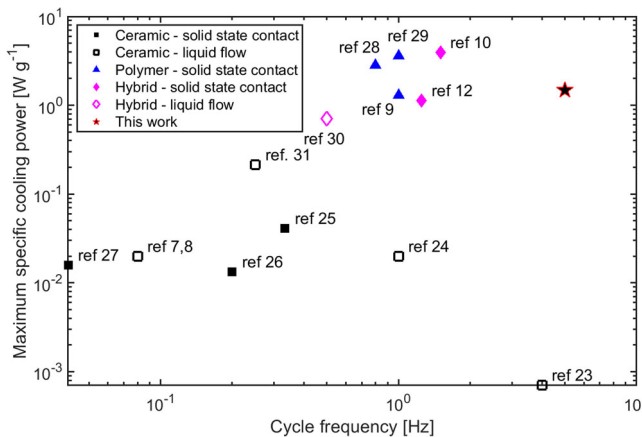

**Fig. 5 | Maximum specific cooling power per active mass of electrocaloric prototype operated at their optimum cycle frequency f.** The empty symbols refer to all prototype, which use fluids for the heat transport, the filled symbols refer to all prototype which use solid state contact for heat transfer. This work, which uses evaporation and condensation as a heat transfer mechanism is shown as a star[22–31].

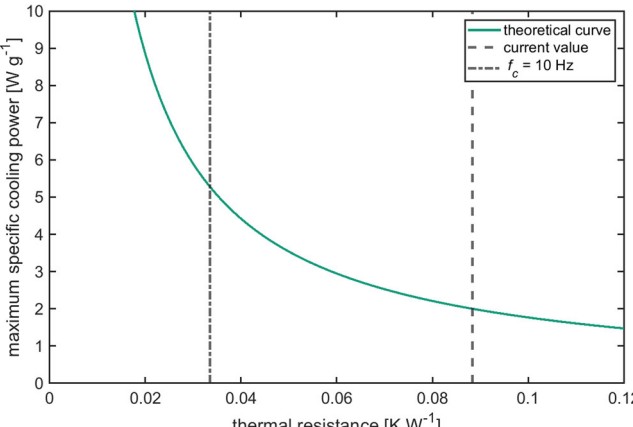

**Fig. 6 | The dependence of the maximum specific cooling power on the thermal resistance, according to equation (3).** The fitted thermal resistance of the device is shown as a dashed line and the thermal resistance needed to achieve a cut-off frequency of 10 Hz is shown as a dashed-dotted line.

Supplementary Fig. 8 and Supplementary Table 1). This demonstrates that with our approach compact EC-devices with considerable cooling power can be built.

While with this concept very good values regarding the cooling power and cooling power density were achieved, the generated temperature span of 0.4 K is still rather small and only accounts for about 25% of the adiabatic temperature change of the electrocaloric components. The main reason for this reduced temperature lift is due to the wick system implemented in the EC-segment for thermal stabilization (see Supplementary Text: Thermal stabilization of EC-components). This wick introduces additional fluid into the system. If too much additional fluid is added to the segment, this acts as an additional thermal capacity. This in combination with a relatively small amount of electrocaloric components used in this segment reduces the temperature lift of the system. This aspect will be improved in future systems e.g. by increasing the number of EC-components and optimize the encapsulation of the wick. This will enhance the ratio of active EC-components to dead thermal mass in the EC-segment and thus increase the temperature span per segment.

## Outlook
We showed that the AEH concept can reach high specific cooling powers, because it can work with higher frequencies than other EC-prototypes using evaporation and condensation as heat transfer mechanism.

The actual critical frequency is lower than the one reached in a magnetocaloric active heat pipe[15], because of the geometry of the components, which results in a high thermal resistance of the components.

Optimizing the geometry of the components, so that a high ratio of active surface to thickness is reached, can lower thermal resistance, and thus strongly improve performance.

Figure 6 shows how the maximum specific cooling power $\dot{q}_{max}$ depends on the thermal resistance $R_{th}$ of the segment (see equation (3)). If the components are made thinner, so that the thermal resistance is reduced to roughly 0.03 K W$^{-1}$, then the critical frequency would be around 10 Hz and the maximum specific cooling power would be roughly 5 W g$^{-1}$.

This would outperform even the best electrocaloric prototype working with polymers or hybrid materials.

It is important to note that the AEH can also work with polymers or hybrid materials, which opens the path to even higher specific cooling powers to more than 10 W/g.

## Conclusion
With the active electrocaloric heatpipe we present an approach for building an electrocaloric heat pump, where the heat transfer is realized by

evaporation and condensation. This allows for higher cycle frequencies, leading to a specific cooling power one order of magnitude larger than other ceramic EC prototype. The performance of the actual system is limited by the heat flow through the components. Components with higher thermal conductivity or an optimized geometry can strongly enhance the performance. The concept of the AEH can also be realized with polymer or hybrid components, where even higher specific cooling powers are expected.

Using latent heat transfer in electrocaloric systems opens the path to high specific cooling powers, which are needed for applications. Environmentally friendly electrocaloric systems, which can compete with vapor-compression based technology are at sight.

## Methods
### Electrocaloric material
The electrocaloric components used are PbSc$_{0.5}$Ta$_{0.5}$O$_3$ (PST) multi-layer capacitors provided by Murata Manufacturing Company. A single PST multilayer capacitors has a length of 10.4 mm, a width of 7.4 mm and a thickness of ~420 μm. The active area of the same components was determined by Nouchogkwe et al.[19] to be 48.7 mm$^2$. The ceramic layer thickness is 38.6 μm and 9 layers are active, which results in an active volume of 16.9 mm$^3$. Given the density of the lead scandium tantalate of 9070 kg m$^{-3}$ determined by Nouchogkwe et al.[21] the active mass is calculated to be 0.15 g.

The adiabatic temperature change at 29 °C was determined to be 1.7 K with an electric field of 5.2 V μm$^{-1}$ applied (see Supplementary Fig. 6). Thin wires are soldered to the EC-components to electrically contact them. The faces with electrical contact are then electrically insulated with HumiSeal 1B51.

### Set up of active electrocaloric heat pipe
A Pfeiffer TSH 071 vacuum pump first evacuates the heat pipe, which is built from standard vacuum components from Novotek. Afterwards the evaporator (which is filled with degassed ethanol), and the condenser reservoir (also filled with degassed ethanol) are opened. This leads to a saturated ethanol steam atmosphere.

The saturated steam atmosphere leads to the correspondence of temperature and pressure according to the vapor pressure curve. The pressures are measured with two CMR371 pressure sensors from Pfeiffer and recorded with a Sefram DAS701. From the pressures, the temperatures are calculated according to the fluid parameters from the database of the National Institute of Standards and Technology (NIST).

The condenser is held at constant temperature of 29 °C with a Mini-chiller 600 H OLÉ from Huber to be in a temperature range, where the ECM shows high adiabatic temperature change.

**Article**

The environment was kept at a temperature of 31 °C to make sure the fluid condenses in the condenser and not elsewhere.

The voltage was applied to the EC-components with a SM 1500 CP 30 power source from Delta Elektronika, which was controlled with a Lab-VIEW software.

The applied voltage was also recorded with the Sefram DAS701.

The cycle frequency was controlled with a LabVIEW® software, which controls the power source and verified from the time evolution of the voltage signal.

By application of a sinusoidal voltage to the EC-components with peak value of 200 V (5.2 V $\mu m^{-1}$) a heat flow was generated from the evaporator to the condenser, leading to a temperature span. When a steady state was reached, heat was applied (like shown in Fig. 2) with a heating wire, which was attached to the evaporator. The heating power (voltage and current) was recorded with a HMC8015 power analyzer from Rohde & Schwarz.

While working, there is a stream of fluid from the evaporator to the condenser. The wick, which is self-built from cleanroom wipes (Cleanmaster Protex 1 from Rohde Clean GmbH), transports liquid from the condenser via the local fluid back to the evaporator.

Because the environment is warmer than the condenser, the evaporator is also warmer than the condenser, so the pressure of the evaporator without working of EC-components is in average 0.48 mbar (roughly 0.09 K) higher than the pressure of the condenser. So, the system has to cool down the evaporator before reaching zero temperature difference between condenser and evaporator. To consider this pressure offset, the pressure difference between condenser and evaporator before each measurement is added to the recorded condenser temperatures. Thus, the real temperature drop of the evaporator is obtained.

### Calculation of specific cooling power

The steady state temperature difference is plotted against the applied heating power and a linear regression is made (Fig. 3). The maximum cooling power at zero temperature span is obtained as the interception of the linear regression with the x-axis (temperature difference = 0). The maximum cooling power is divided by the active mass of the EC-components to get the specific maximum cooling power. This procedure is repeated for different cycle frequencies for the applied voltage from 0.3 to 5 Hz. Then the specific maximum cooling powers are plotted against the cycle frequencies and are analyzed with the equations provided by Hess et al.[18].

### Quantification of measurement uncertainties

**Temperature span $\sigma_{\Delta T}$.** The uncertainty of the pressure sensors is given as 0.15% of the measured value. The pressure values are around 100 mbar, so the uncertainty is 0.15 mbar.

The uncertainty of the Sefram DAS701 is given as 0.1% of the range. The range was 20 mbar, so the uncertainty is 0.02 mbar.

With gaussian error propagation the uncertainty in pressure is calculated and translated into an uncertainty in temperature according to the vapor pressure curve (from NIST) and results in an uncertainty of 0.03 K. This is the uncertainty in temperature for the evaporator and the condenser individually.

There is an additional uncertainty from the pressure offset correction, which is 0.02 K. For the temperature difference between condenser and evaporator it follows with gaussian error propagation a total uncertainty of 0.05 K.

**Specific cooling power $\dot{q}$.** The uncertainty of the specific cooling power $\dot{q}(\Delta T = 0)$ is calculated from the interception of the lower and upper 95 % confidence interval of the linear regression with the x-axis ($\Delta T = 0$). This is graphically shown in Supplementary Fig. 7.

### Data availability

The data that support the findings of this study are available from the corresponding author upon request.

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

## Acknowledgements
This work was supported by the Fraunhofer Society through the Fraunhofer Lighthouse Project ElKaWe. We thank H. Kuramoto and K. Sasaki for their assistance in fabricating the MLCs. We thank Roland Binninger and Manuel Näher for help in designing the EC-segment.

## Author contributions
Conceptualization: K.B., D.B., J.M., P.C., F.K., S.M. Resources: S.H. Methodology: P.C., J.M., S.H., D.L. Investigation: P.C., J.M., D.L. Funding acquisition: K.B., O.S.W., D.B., S.M. Project administration: D.B., K.B. Supervision: O.S.W., K.B. Writing – original draft: J.M., K.B., D.B. Writing – review & editing: S.M., J.M., K.B., D.B., F.K.

## Funding

## Competing interests
The authors declare the following competing interests: The PST-samples integrated in the described system were provided by Murata Manufacturing Co. The system concept described in the paper is basis of the patent WO2016008732A1 (Applicant: Fraunhofer Institute for Physical Measurement Techniques IPM, Inventors: Kilian Bartholomé and Jan König, Application Status: National Phase Entry).

## Additional information
**Supplementary information** The online version contains Supplementary Material available at https://doi.org/10.1038/s44172-024-00199-z.

