## [Peer Review File · Communications Engineering]

Reviewers' comments:

Reviewer #1 (Remarks to the Author):

The authors propose to use a heat pipe approach to transfer the heat generated from the EC-material. With a heat transfer through condensation and evaporation of ethanol, they mentioned a working frequency around 5 Hz and a corresponding specific cooling power of 1.5W/g, 36 times higher than the state of the art. The concept is interesting, however the 1.5W/g has to be taken with a pinch of salt as in the comparison the temperature span is put aside.

To understand the working principle in detail is challenging. I struggle trying to understand the heat and mass balance in the system. From my point of view, understand the system working principle is too challenging from the document. With this active caloric heatpipe, authors has done similar experiment with magnetocaloric and elastocaloric material as shown in reference 15 and 16. Nonetheless, the working principle has to be explained in more detail. It is hard for example to understand the pressure all along the device (see fig S2)? In addition, is the equation (4) in supplementary correct?

Thereafter, authors studies limitation of the power regarding the cycle frequency or of heat transfer limitation within the components. Author shows the key role of the EC-component thickness through the cutoff frequency. This part is more natural. However, is it easy to use/make thinner EC?

Finally, a comparison of the specific cooling power, without taking into account the temperature span, is not relevant. At least to avoid confusion, they have to be mentioned. Fig. S9 has to be in the main document and comment either on the temperature difference and on specific cooling power. The video shows a cascaded system, why is here only a single stage? Is a cascade feasible? Do it rise new challenges? Do the check valve have some frequency limitation? Authors wrote "When working with a liquid fluid, the thermal coupling of the EC material to the fluid is good. But the fluid velocities have to be low to prevent turbulences, which increases the hydraulic losses. deteriorate the established temperature gradient and system efficiency" what about the fluid motion in their device?

Reviewer #2 (Remarks to the Author):

The author proposed a new structure combining solid-state electrocaloric material with condensation and evaporation of heat transfer fluid, and the power density is successfully

enhanced. However, the performance is still not superior compared to other works. This paper could be published after a major revision addressing the following questions.

1. As shown in Figure 3, the specific cooling power is comparable to polymer and hybrid electrocaloric cooling methods. The author needs to explain what is the advantage of this work compared to those with similar specific cooling power.
2. As shown in Figure S4 and S8, the temperature lift of this device is $<0.5\text{K}$, which is lower than the adiabatic temperature of the material itself. The author needs to explain the reason and propose a way to overcome this drawback.
3. How much does the condenser system contribute to the cooling effect? The author should conduct a comparison experiment with electrocaloric effect off and measure what is the temperature lift.
4. The temperature profile data shown in Figure S8 is essential to the paper. It should be moved to the main figures.

Reviewer #1 (Remarks to the Author):

1. The authors propose to use a heat pipe approach to transfer the heat generated from the EC-material. With a heat transfer through condensation and evaporation of ethanol, they mentioned a working frequency around 5 Hz and a corresponding specific cooling power of 1.5W/g, 36 times higher than the state of the art. The concept is interesting, however the 1.5W/g has to be taken with a pinch of salt as in the comparison the temperature span is put aside.

Thank you for this relevant feedback. We agree that of course the temperature span is another relevant performance figure of a caloric system. However, at this point we wanted to demonstrate the effect of our system approach to increase the power density by enabling higher cycle frequencies. The EC-segment was not yet optimized with respect to the temperature span. The wick system implemented together with the additional heat transfer fluid introduces an additional thermal capacity to the EC-segment which has a detrimental effect on the attainable dT between the evaporator and the condenser. This aspect will be improved in future systems to increase the temperature span per segment.

To account for this, we inserted the following paragraph to the section “comparison to literature” on page 8:

“While with this concept very good values regarding the cooling power and cooling power density were achieved, the generated temperature span of 0.4 K is still rather small and only accounts for about 25 % of the adiabatic temperature change of the electrocaloric components. The main reason for this reduced temperature lift is due to the wick system implemented in the EC-segment for thermal stabilization (see supplementary text: Thermal stabilization of EC-components). This wick introduces additional fluid into the system. If too much additional fluid is added to the segment, this acts as an additional thermal capacity. This in combination with a relatively small amount of electrocaloric components used in this segment reduces the temperature lift of the system. This aspect will be improved in future systems e. g. by increasing the number of EC-components and optimize the encapsulation of the wick. This will enhance the ratio of active EC-components to dead thermal mass in the EC-segment and thus increase the temperature span per segment.”

Furthermore, we also included the dT_{\max} of each system in Supplementary Table 1.

2. To understand the working principle in detail is challenging. I struggle trying to understand the heat and mass balance in the system. From my point of view, understand the system working principle is too challenging from the document. With this active caloric heatpipe, authors has done similar experiment with magnetocaloric and elastocaloric material as shown in reference 15 and 16. Nonetheless, the working principle has to be explained in more detail. It is hard for example to understand the pressure all along the device (see fig S2)? In addition, is the equation (4) in supplementary correct?

We agree, the working principle is challenging to understand. We therefore added a more detailed description in the main text:

“Whenever an electric field is applied to the EC-components, their temperature increases due to the electrocaloric effect. Thereby the liquid fluid evaporates from the surface of the EC-components, and the pressure inside the segment increases. This leads to an opening of the check valve and a gaseous fluid flow to the hot side, transferring heat from the EC-components to the hot side (condenser). When subsequently the electric field is turned off, the EC-components cool down and gaseous fluid

condenses on the components. This leads to a decrease in pressure in the segment, opening the check valve to the cold side and letting gaseous fluid enter from the evaporator. Thus, by an alternating electric field, a unidirectional gas flow from the cold side to the hot side occurs, whereby heat is pumped.”

We also adapted the explanation of the imbalance of thermal energy and fluid flow in an electrocaloric heat pipe without local fluid return in the supplement in order to better motivate the implementation of the local fluid return.

Regarding former equation 4: This equation is correct. A pressure difference is built up between the condenser and evaporator during operation. Pressure and temperature are directly coupled in a heat pipe (saturated vapor curve).

3. Thereafter, authors studies limitation of the power regarding the cycle frequency or of heat transfer limitation within the components. Author shows the key role of the EC-component thickness through the cutoff frequency. This part is more natural. However, is it easy to use/make thinner EC?

Thank you for the question:

Yes, it is possible to decrease the thickness of the EC-components by further decreasing the thickness of each PST layer or/and using fewer layers. Currently 11 layers are used (9 active, 2 inactive). Although it must be pointed out that the reduction of the thickness has limitations as the handling of the MLCs becomes increasingly difficult due to the brittleness of ceramics. However, obtaining MLC thickness of around 220 to 240 μm is realistic using current production methods and such MLCs would be possible to handle when building a system.

As pointed out in the main text, the cut-off frequency of the system is directly coupled to the thickness of the EC-components. By reducing the thickness of the MLCs to the above-mentioned value a higher cut-off frequency would be obtainable, as the thermal diffusion time is largely reduced. This would lead to a significant reduction in the overall thermal resistance of the system and lead to a cut-off frequency around 10 Hz as shown in Fig. 4.

4. Finally, a comparison of the specific cooling power, without taking into account the temperature span, is not relevant. At least to avoid confusion, they have to be mentioned. Fig. S9 has to be in the main document and comment either on the temperature difference and on specific cooling power. The video shows a cascaded system, why is here only a single stage? Is a cascade feasible? Do it rise new challenges? Do the check valve have some frequency limitation? Authors wrote “When working with a liquid fluid, the thermal coupling of the EC material to the fluid is good. But the fluid velocities have to be low to prevent turbulences, which increases the hydraulic losses deteriorate the established temperature gradient and system efficiency” what about the fluid motion in their device?

Thank you for this important remark:

- Please see our response to comment 1. We believe the comparison is still relevant as it shows the potential of the system, which was the main goal of this work. We concur however, that the dT of the system still needs to be improved significantly. For clarification we added a paragraph to the main text (see comment 1)
- As suggested, we have included Fig. S9 in the main body of the text and included remarks on the achieved dT in the system and the current limitations.

- The system in this work is based on a single EC-segment including check valves and the electrocaloric materials. The EC-segment is the basic building block of a cascaded heat pipe system (as depicted in Fig. 1 and the video). To obtain a temperature difference higher than the dT_{ad} of the individual EC-components more than one EC-segment must be operated in series. Nevertheless, to show the effect of an increase power density, a cascaded system, which is more complex to build, is not necessary. Building a cascaded system is feasible and will be subject of future work in which we plan to demonstrate the working of a cascaded electrocaloric system. To reduce the complexity of the setup in this work we aimed at building the simplest form of an electrocaloric heat pipe system.
- Regarding the hydraulic losses: in electrocaloric systems, where the heat transfer is accomplished by pumping a liquid fluid, relatively large fluid fluxes have to be realized in order to achieve large cooling powers. This is often accompanied by large pressure drops and thus by large hydraulic losses. When evaporation and condensation is used as heat transfer, very small amounts of fluid are sufficient to transfer the heat due to the very large enthalpy of evaporation/condensation. Thus, the fluid fluxes and thereby pressure drops are orders of magnitude smaller than compared to liquid-based heat transfer.

Reviewer #2 (Remarks to the Author):

The author proposed a new structure combining solid-state electrocaloric material with condensation and evaporation of heat transfer fluid, and the power density is successfully enhanced. However, the performance is still not superior compared to other works. This paper could be published after a major revision addressing the following questions.

1. As shown in Figure 3, the specific cooling power is comparable to polymer and hybrid electrocaloric cooling methods. The author needs to explain what is the advantage of this work compared to those with similar specific cooling power.

Thank you for this good remark. In this work we aimed at proving the concept of latent heat transfer for electrocaloric components in order to attain large cycle frequencies and thereby large cooling power densities. This we have successfully demonstrated for ceramic components. Since the polymer- and hybrid materials can withstand higher electrical fields and thus show higher entropy changes per mass than ceramic EC-material, their integration in a latent-heat-transfer-based system promises even larger cooling power densities of more than 10 W/g. We added this statement in the outlook.

2. As shown in Figure S4 and S8, the temperature lift of this device is $<0.5\text{K}$, which is lower than the adiabatic temperature of the material itself. The author needs to explain the reason and propose a way to overcome this drawback.

As you have pointed out the temperature lift of our system is currently below the dT_{ad} of the used PST MLC. This is a result of the following current design limitation:

- Small amount of active mass in the EC-segment
Currently, due to limited availability of EC-components, only very small amounts for EC-material has been used in the EC-segment, and these EC-components are only about 50% active, the remaining parts are inactive. However, the lower the active mass is in the system, the lower is the available thermal energy to build up a temperature lift and to pump heat. Since there exist parasitic heat fluxes (e.g. insufficient isolation from the environment, dead surfaces in the segment, dead thermal mass in the segment) in the system the influence in the current system is proportionally large
- Please also see the answer for reviewer 1 under 1: The EC-segment with the electrocaloric components was not yet optimized for reaching a high dT . The wick system implemented together with the additional heat transfer fluid introduces “dead” thermal mass to the EC-segment which has a detrimental effect on the attainable dT between the evaporator and the condenser. Furthermore, a neglectable amount of the observed temperature losses result from the used check valves.
- We have included a paragraph explaining the current limitations and how to overcome them in paragraph “comparison to literature” on page 8.

3. How much does the condenser system contribute to the cooling effect? The author should conduct a comparison experiment with electrocaloric effect off and measure what is the temperature lift.

As can be seen in former Fig. S8, now figure 2 the temperature of the condenser and evaporator start out at the same temperature before the system is started, and an electric field is applied in form of a

sine wave. The temperature of the evaporator starts to decrease once the system is started. From this data we conclude that there is no significant cooling effect when the voltage source is off.

Fig. S8.

Temporal evolution of the condenser and evaporator temperatures of our system. After application of a sinusoidal voltage to the electrocaloric components a temperature difference establishes (no load condition). When the evaporator is heated, the established temperature difference reduces. For each heating power the steady state temperature difference is recorded (corresponding to one data point in fig. S9).

4. The temperature profile data shown in Figure S8 is essential to the paper. It should be moved to the main figures.

Thank you for your input. Due to the above-mentioned comment, we moved the Fig. S8 in the main body of the paper and explained it in the text.

Further changes to manuscript:

With the new paper of Li et al. 2023 being published in November 2023, we added this paper to the references (ref. 32) as well as to the figures comparing our system to systems in the literature (figure 5 and supplementary figure 8)

REVIEWERS' COMMENTS:

Reviewer #1 (Remarks to the Author):

Authors have answered comments and questions. An the paper is ready for publication. For future work, I am curious to understand more the working principle. And from my point of view I still find explanations of the working principle limited to a description of the mechanisms involved rather than to physical modeling.

Reviewer #2 (Remarks to the Author):

The authors has addressed my concerns, and this paper could be considered for publication in Communications Engineering.